# Initiating gender-affirming hormones for transgender and non-binary people: A qualitative study of providers' perspectives on requiring mental health evaluations

Daphna Stroumsa[1,2]*, Leah A. Minadeo[3], Mariam Maksutova[4], Molly B. Moravek[1], Rob Stephenson[5,6], Paul N. Pfeiffer[2,7], Justine P. Wu[2,8]

1 Department of Obstetrics and Gynecology, University of Michigan Medical School, Ann Arbor, Michigan, United States of America, 2 Institute for Healthcare Policy & Innovation, University of Michigan, Ann Arbor, Michigan, United States of America, 3 School of Information Sciences, Wayne State University, Detroit, Michigan, United States of America, 4 Medical School, University of Michigan, Ann Arbor, MI, United States of America, 5 Department of Systems, Populations and Leadership, School of Nursing, University of Michigan, Ann Arbor, Michigan, United States of America, 6 The Center for Sexuality and Health Disparities, University of Michigan, Ann Arbor, Michigan, United States of America, 7 Department of Psychiatry, University of Michigan Medical School, Ann Arbor, Michigan, United States of America, 8 Department of Family Medicine, University of Michigan Medical School, Ann Arbor, Michigan, United States of America

* daphnast@med.umich.edu

**Data Availability Statement:** Due to the nature of the qualitative data, and the participants coming

## Abstract

### Objective

Gender-affirming hormones (GAH)—the use of sex hormones to induce desired secondary sex characteristics in transgender individuals—is vital healthcare for many transgender people. Among prescribers of GAH, there is debate regarding the value of a universal requirement for an evaluation by a mental health provider prior to GAH initiation. The purpose of this qualitative study was to describe the range of attitudes and approaches to mental health evaluation among GAH providers in the United States. We analyzed the providers' attitudes and base our recommendations on this analysis.

### Methods

We conducted semi-structured interviews with 18 healthcare providers who prescribe GAH across the United States. Participants were purposefully recruited using professional networks and snowball sampling to include those who require mental health evaluation and those who do not. We adapted domains from the Theoretical Domains Framework—a framework for understanding influences on health professional behavior—to inform the interviews and analysis. Guided by these domains, we iteratively coded text and identified theoretical relationships among the categories.

### Results

While some felt a universal requirement for mental health "clearance" was necessary for the identification of appropriate candidates for GAH, others described this requirement as a

from a small, well-connected pool of providers who prescribe gender-affirming care in the U.S., it is not possible to sufficiently de-identify the data to make it anonymous; nor have we obtained permission from participants when they granted their informed consent to the study. Per your policy, we have shared extensive excerpts within the manuscript itself— more so in the revision than in the original document. In addition, we have shared aggregated data in Table 1. Beyond this, following the HHS guidelines below, we are unable to de-identify the data, and hence would be in violation of our commitment to participants by sharing it. De-Identification https://www.hhs.gov/hipaa/for-professionals/privacy/special-topics/de-identification/index.html#safeharborguidance Identifying Characteristic https://www.hhs.gov/hipaa/for-professionals/privacy/special-topics/de-identification/index.html#_edn1 We believe that these criteria are applicable to our study, given the small and tight-knit community of gender-affirming hormone providers in the US. Upon request and subsequent review by our institution's data sharing team, we can make the data available to interested and qualified researchers under the appropriate Data Use Agreement for the identifiable nature of the data. The contact information is below. University of Michigan Medical School Institutional Review Board (IRBMED) 2800 Plymouth Road, Building 520, Suite 3214 Ann Arbor, MI 48109-2800 phone (734) 763 4768 fax (734) 763 9603 irbmed@umich.edu Reference: HUM00149212.

**Funding:** DS: Blue Cross Blue Shield of Michigan Foundation, Physician Investigator Research Award (#002761.PIRAP): Models for Initiation Decision for Gender Affirming Hormones: Patient and Provider Perspectives. https://www.bcbsm.com/foundation/grant-programs/investigator-initiated-research.html DS: National Institutes for Minority Health and Health Disparities grant #K23 MD016950." JPW: University of Michigan Institute for Research on Women and Gender, Faculty Seed Grant (no grant #): Transgender Health and Empowerment. https://irwg.umich.edu/funding/irwg-faculty-seed-grant The funders had no role in study design, data collection and analysis, decision to publish, or preparation of the manuscript.

**Competing interests:** The authors have declared that no competing interests exist.

form of "gatekeeping" that limits access to care. Themes we identified included providers' ability to ascertain gender identity; concern about mental illness; GAH provider and mental health provider expertise; and provider roles. All providers appreciated the potential advantages of mental health support during GAH treatment.

## Conclusion

Providers in our study practice on a continuum of care rather than adhering to strict rules about the requirement for mental health evaluation prior to GAH treatment. Where they fall on this continuum is influenced primarily by their perceptions of transgender identity and transition, and their interpretation of risk for significant mental illness and its association with transness. Providers who required universal evaluation by a mental health professional tended to hold essentialist, medicalized, and binary ideas of gender and transness.

## Introduction

In the United States, 78% of approximately 1.4 million transgender adults [1]—those whose gender identity differs from the sex they were assigned at birth—seek or receive gender-affirming hormones (GAH) [2]. There is broad agreement in the medical community that GAH is an essential, medically necessary health service [3–6] that can improve the physical and mental health and well-being of transgender individuals [7, 8]. Despite this, there is currently no consensus among experts and healthcare providers regarding the need for evaluation by a mental health specialist prior to GAH treatment [9–12].

The World Professional Association of Transgender Health (WPATH) Standards of Care, first published in 1979 and now on its seventh iteration [13], recommend universal evaluation by a mental health provider (MHP) and formal documentation that the patient is an appropriate candidate for GAH with a diagnosis of "gender dysphoria" prior to initiating care [14]. Gender dysphoria, defined in the 2013 version of the Diagnostic and Statistical Manual of Mental Disorders, 5th Edition: DSM-5 [15], refers to emotional distress caused by the incongruence between a person's sex assigned at birth and their gender identity. WPATH recommends that this documentation, usually in the form of a letter, includes results of a psychosocial assessment and any psychiatric diagnoses. This letter should document that the criteria for hormone therapy have been met and the clinical rationale for hormone therapy, as well as a statement indicating that informed consent has been obtained from the patient. Though systematically developed by an organization devoted to the treatment of transgender individuals, these guidelines reflected medical convention at the time to treat "transsexualism" as a psychiatric pathology [DSM-III] by a society that consisted primarily of MHPs at the time. The Endocrine Society has issued recommendations similar to WPATH in its guidelines [6]. Though widely utilized, experts and advocates have recently scrutinized the WPATH approach as paternalistic, restrictive, and without scientific evidence for benefit [12, 16–21]. Although the WPATH standards of care [14] leave room for alternative approaches, the recommendation for an MHP letter prior to hormone initiation is commonly referred to as "the WPATH model" by patients and providers alike.

While some GAH providers adhere to the WPATH recommendation, others do not consistently require mental health evaluation and instead use a process commonly referred to as "informed consent" (henceforth, IC) for GAH [16]. The IC model emerged in the 2000s in an

attempt to depathologize transgender identity and increase access to gender-affirming care broadly and GAH specifically [12, 16, 21]. This approach has been loosely defined by the *absence of a requirement* for an MHP letter, distinguishing it from the WPATH model [16]. At its idealized core, the IC model centers the patient as the primary decision-maker and asserts that GAH may be initiated if the patient has the ability to consent and fully understands the potential benefits and risks [13]. The choice of approach is currently a matter of provider preference [11–13, 16]. There is little literature regarding providers' perceptions of the advantages and disadvantages of each approach.

To address this critical gap in knowledge, we aimed to describe the range of experiences and attitudes regarding hormone initiation among GAH providers and to assess the factors that affect these attitudes—including the perceived advantages and disadvantages of each—and to base recommendations for future guidelines on our findings. We utilized the Theoretical Domains Framework [22]—a theory-based approach used to understand the cognitive, social, and environmental influences on healthcare professionals' behavior. In this paper, we use the term "transgender" as a non-exclusive umbrella term, to include people identifying along the spectrum of non-cisgender identities, including non-binary and genderqueer individuals, among others.

## Materials and methods

### Researcher reflexivity

Our interprofessional team consisted of clinicians, researchers, and research staff with collective expertise in qualitative research and the intersection of healthcare for transgender and non-binary people, sexual and reproductive health, mental health, and access to care. Our team consists of cisgender, queer, and non-binary people, of a diversity of genders and sexualities. While some of us have sought a variety of forms of medical gender affirmation (in addition to the casual, everyday gender affirmation that cisgender people receive by society at large), none of us have accessed gender-affirming hormones. This latter fact may well have impacted our understanding of the process and our interpretations. Most of us are white. We are mindful that we occupy a privileged social location as academics and healthcare professionals.

### Study design

In this paper, we present findings from interviews with healthcare providers as part of a larger qualitative study. Eligible participants were healthcare providers (physicians or advanced practice professionals) across the United States who had prescribed GAH to at least 10 patients in the past year. We purposefully sampled participants based on their self-description as using either IC models or WPATH Standards of Care guidelines in initiation of a GAH prescription. We also purposefully sampled participants from a diversity of specialties and geographical areas within the United States. For recruitment, we used snowball sampling initiated from providers within the principal investigator (PI)'s network, as well as from a listserv of GAH providers. A call for participants was placed on the listserv and individual participants were approached via email. The study was approved by the University of Michigan Institutional Review Board. We conducted interviews from May to October of 2019.

A trained research assistant and the PI (LM, DS) conducted semi-structured, qualitative interviews of eligible and verbally consented participants. Prior to the interview, participants completed a demographic and professional background survey. Interviews lasted 30–60 minutes. Because participants were recruited from all regions in the United States, we used remote communication of the participant's choice (Skype or phone). Participants were asked to ensure

that they were in a private space during the interview. The interviews were audio-recorded with the participant's permission and professionally transcribed with personal identifiers omitted. The transcripts were coded and analyzed using Dedoose software. Participants received $50 for completion of the interview.

### Theoretical framework and interviews

The interview guide was constructed using the Theoretical Domains Framework [15], a framework for understanding influences on health professional behavior. To assess barriers and facilitators to GAH care, we explored domains of 1) social/professional role and identity (how does the respondent understand their job and their professional responsibilities?); 2) social influences (how did they come to providing GAH; who trained them?); 3) beliefs about consequences (what outcomes do they anticipate; what experiences shaped these beliefs?); 4) motivation and goal (what are their intentions; how do they set their goals?); 5) knowledge and skills; and 6) environmental context and resources (what barriers and facilitators do they face in prescribing GAH?). Participants were asked to describe their experiences prescribing GAH, the path they took to becoming a GAH provider, and the process they use when initiating hormones—including any mental health evaluations and interaction with other professionals. The interview guide was pilot tested by two of the researchers who are also physicians and GAH providers (DS, JW).

**Data analysis.**   We analyzed the coded interviews using thematic analysis. Inductive methodology was used to develop codes [16]. The first author (DS) and coders (LM, MM) familiarized themselves with the dataset by listening to the audio recordings and reading the transcripts and looking for repeated topics or codes. Two team members (DS, LM) jointly coded the initial interviews using Dedoose software. We developed a list of relevant codes, iteratively adjusting the list as we individually coded the remainder of interviews. The research team met weekly to further analyze the narratives. We used a matrix worksheet in Excel (version 16.4) to compare and contrast codes between providers by their stated approach (WPATH vs. IC) and by their practice. Using thematic analysis [23], the team combined the codes into higher-level categories and identified theoretical relationships among the categories.

## Results

We interviewed 18 providers, of whom 11 self-described as using "informed consent" and seven self-identified as following "the WPATH model" or another approach. The majority of providers were white (94%) and primary care providers (61%). Eleven (61%) participants identified as cisgender women, five (28%) as cisgender men, and two (11%) as transgender or nonbinary (Table 1).

Providers described a variety of approaches to prescribing hormones (Fig 1). Their practices included requiring letters from specific MHPs; requiring evaluation by an in-house MHP; requiring letters that then underwent additional evaluation by an in-house MHP; or selective referral for MHP evaluation or treatment, with or without a requirement for a letter prior to hormone initiation in those select cases. Of those who did not universally require an MHP evaluation, some had a low threshold for referring patients for MHP evaluation, while others did so rarely. Some required a signed consent form, while others did not.

Providers attributed their decision about MHP involvement to a variety of factors. Themes ranged from perceptions regarding practical concerns with individual patients ("Are they insured or not? Whose insurance are they on? Are they out? . . . I think the biggest barrier is the financial piece and the safety and security") to broader attitudes regarding gender identity

**Table 1. Characteristics of respondents.**

| Characteristic | All Respondents (n = 18)[a] |
|---|---|
| Age, years[b] | 44.5 (39.3, 55.5) |
| Age Group | |
| 30–39 | 4 (22.2) |
| 40–49 | 9 (50.0) |
| 50–69 | 5 (27.8) |
| Race | |
| White | 17 (94.4) |
| Asian | 1 (5.6) |
| Region | |
| Midwest | 6 (33.3) |
| Northeast | 7 (38.9) |
| South | 1 (5.6) |
| Southeast | 1 (5.6) |
| West | 3 (16.7) |
| Gender Identity | |
| Cisgender man | 5 (27.8) |
| Cisgender woman | 11 (61.1) |
| Non-binary | 1 (5.6) |
| Transgender man | 1 (5.6) |
| Sex Assigned at Birth | |
| Female | 13 (72.2) |
| Male | 5 (27.8) |
| Sexual Orientation | |
| Gay | 1 (5.6) |
| Heterosexual | 9 (50.0) |
| Lesbian | 2 (11.1) |
| Pansexual | 1 (5.6) |
| Queer | 4 (22.2) |
| No answer | 1 (5.6) |
| Practice Setting | |
| Rural | 0 (0.0) |
| Suburban | 1 (5.6) |
| Urban | 11 (61.1) |
| Multiple/Other | 6 (33.3) |
| Screening Approach | |
| Informed Consent | 11 (61.1) |
| WPATH | 4 (22.2) |
| Other | 3 (16.7) |
| Years in Practice[b] | 12.5 (5.75, 23.0) |
| Specialty | |
| Family Medicine | 10 (55.6) |
| Family Nurse Practitioner | 1 (5.6) |
| Internal Medicine | 5 (27.8) |
| Obstetrics and Gynecology | 1 (5.6) |
| Pediatrics | 1 (5.6) |
| Patients on Gender-Affirming Hormones | |
| ≥100 | 9 (50.0) |

(*Continued*)

**Table 1.** (Continued)

| Characteristic | All Respondents (n = 18)[a] |
|---|---|
| 51–99 | 2 (11.1) |
| 21–50 | 3 (16.7) |
| 1–20 | 3 (16.7) |
| 0 | 1 (5.6) |

WPATH, World Professional Association of Transgender Health

[a]Data presented as n (%) unless otherwise noted

[b]Median (interquartile range)

("Some people do have gender dysphoria but not everybody is dysphoric about just being who they are"; "I think we are sometimes too rigid about how we define gender"). All providers considered the social challenges related to both social and medical transition. They acknowledged the potential social, emotional, and financial implications of potential rejections and importance of social networks in supporting a trans person as they transitioned. These themes were present in our interviews with providers regardless of their approach to hormone initiation. However, there were some distinct differences by approach. In the following sections we review the main themes identified and analyze them according to provider approach to MHP involvement.

## Providers who require MHP evaluation

In the following section we present the themes among providers who strictly required an MHP evaluation prior to GAH initiation. Within this group, we identify the following themes: a) the importance of MHP consultation to ascertain a diagnosis of gender dysphoria; b) concern about unstable or undetected mental health issues; and c) the provider's own confidence in addressing these issues. We detail each of these themes below. These and other advantages and disadvantages of involving an MHP are summarized in Fig 2.

## Ability to ascertain a diagnosis of gender dysphoria and the need for mediation by an MHP

Some of the providers who required a mental health evaluation relied on an MHP for a diagnosis of gender dysphoria or gender incongruence.

> Provider #13: "I don't like to get into the space of making a diagnosis in terms of gender dysphoria. . .because I don't feel like I have that expertise and I don't really have that time to go through initial evaluations. So, I do lean on my mental health colleagues to make all those diagnoses."

In this statement, the provider describes their rationale for outsourcing of the diagnostic evaluation process for what they consider to be the condition that they will then treat. The same provider acknowledged that they tend to believe a patient's self-identification as transgender, but nevertheless require this self-identification to be confirmed by an MHP.

> Provider #13: "there are a lot of people, you know, who are transgender who just do not think the patients require psychiatric clearance. They believe that if somebody comes in and says, you know, I have been the other gender my whole life–you know, and I agree with that."

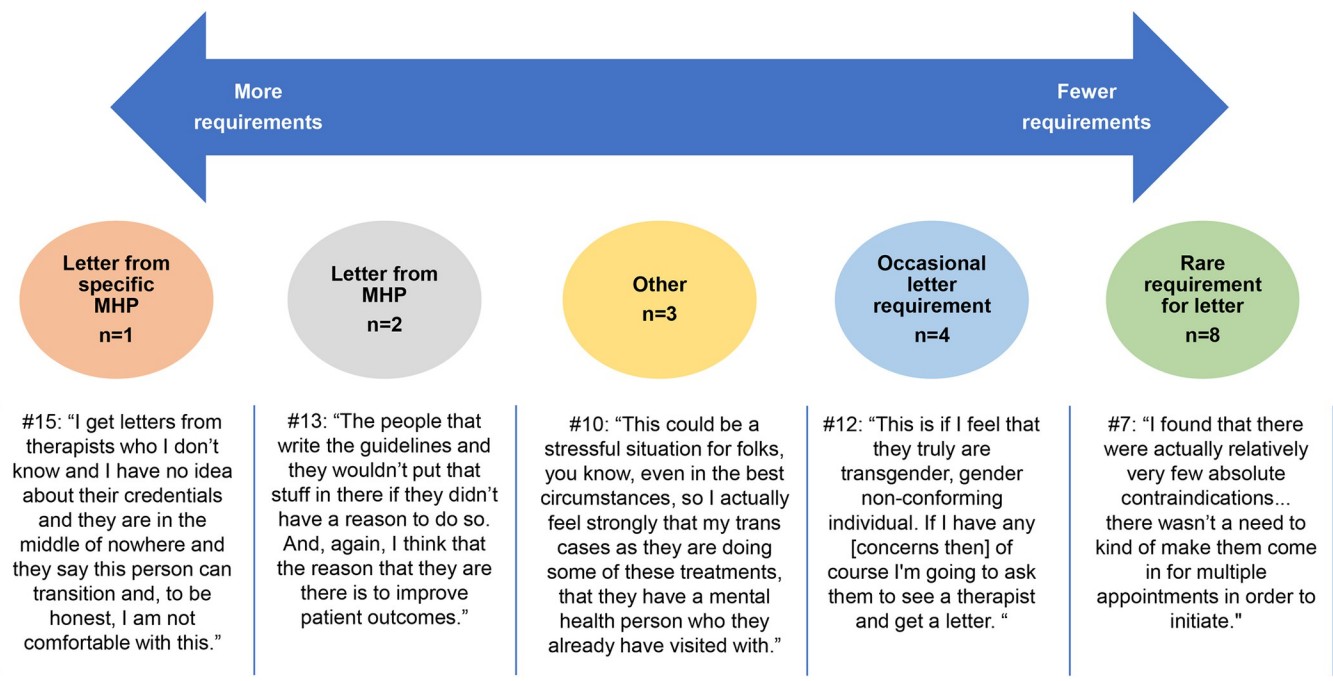

**Fig 1. Providers' perspectives on the role of a mental health provider.** MHP, mental health provider.

This provider acknowledges that in principle, they agree that transness is best attested by the person themselves. Yet, there is an apparent discrepancy between that claim and the provider's insistence on having the patient's identity verified by an MHP. Despite their stated agreement with the principle of believing a patient about their identity, ultimately, it is disbelief or doubt in the patient's transness that determines this provider's course of action.

The provider here also reveals a binary view of gender ("the other gender"). Several providers expressed discomfort with people who identify as non-binary—worrying that such patients were uncertain about their own identity, or that their "true" identity was difficult to ascertain, and thus needed further evaluation. When encountering such patients, they might refer them to a counselor with a particularly rigorous evaluation process (for example, someone who will only write a letter after establishing a prolonged relationship with the patient over several months). As an example, provider #14 stated:

> "Someone that is more gender fluid, maybe leaning towards maybe trans male or trans female so they are, I'm not, like, 'clearly this is what's happening,' I have some question about where they are on the sort of spectrum."

Similarly, provider #15 recounted responding to a patient who stated their goal was to have a queer gender expression by referring them to a psychiatrist:

> "This was a woman that wanted to take a little bit of testosterone. She didn't want to transition to man, she just wanted to be a little bit more masculine... I referred her to psych... she felt like–she wanted to be in between. And there are people who are in between due to real hormonal problems. I felt like I couldn't contribute to that difficulty and treatment."

Both providers (#14 and #15) display fundamental misunderstandings of non-binary and genderqueer identities or of ways of being trans that didn't match up with their expectations

| Advantages | | | Disadvantages | | |
|---|---|---|---|---|---|
| **Universal Screening** | *Diagnosis of gender dysphoria* | "We don't know what causes transsexualism, therefore, we don't know who really should be transitioned."<br><br>"What I do require is that there is an established diagnosis… of gender dysphoria." | *Creates barriers to care (financial, insurance, delay in care, lack of access)* | "I get worried that if we put in too many barriers, those people are going to get lost in care." | |
| | *Screening for severe mental illness* | "Psych service is really just to rule out any underlying problems."<br><br>"I still want that extra help from a more knowledgeable professional if their mood seems very unstable, if they are suicidal or something like that, you know, is this a good time to start therapy?" | *Decreases trust and honesty* | "You are going in with not the ulterior motive of establishing a relationship, just because I need something from you. That's not going to be therapeutic in my mind."<br><br>"Trans people got very good at finding out what was the narrative that would get them a letter. So, what it ended up doing was having people not be honest with the mental health people." | |
| | *Practical reasons: assistance with bureaucracy, in preparation for surgery* | "I usually encourage people who have not engaged with a therapist if they want gender affirming surgery to get engaged so that they can get the letters they need." | *Risk of exposing patients to non-affirming MHPs* | "Especially if they live in a rural area, they may not find anyone supportive that doesn't want to, you know, convert them or something." | |
| | | | *Stigma* | "It's pretty stigmatizing to think that someone couldn't make that decision for themselves." | |
| **Referral of Specific Cases** | *Clarify identity or motivation* | "If there seemed to be something unusual about a person's story that certainly required a letter." | *Concern over capability of MHPs* | "That really doesn't make sense to have someone who doesn't know anything about trans identity say yes, you are not trans or no, you are not trans enough – which did happen quite a bit."<br><br>"I have had several therapists who don't have any experience with the situation… so then, I have to ask that they see another therapist." | |
| | *Support for patients with severe/uncontrolled mental illness* | "Unless someone is unable to consent because they are manic or I am worried about their – I guess their ability to understand the benefits and the risks, then I will talk to their mental health provider or require a letter." | | | |
| | *Clarify ability to give consent (mental/cognitive ability)* | "I don't require a letter from mental health to start hormones… the only time I have had questions – this has come up a couple of times where I just didn't feel the person had the capacity to understand so it's really a capacity issue." | *Futility* | "At the end of the day, if they go to five people and five people say, no, and I meet with them and I'm like, no, you truly have gender dysphoria, like, you meet all the criteria. What am I going to do then? Am I going to not prescribe them hormones even though I fully believe they are or are they going to have to continue to doctor shop until they find that person that agrees with them?" | |
| | *Unrelated need for psychotherapy* | "I'm a poor substitute for a good therapist." | | | |

**Fig 2. Advantages and disadvantages of involving a mental health provider.** MHP, mental health provider.

about sex/gender. Providers were confused when patients with such identities did not fit the providers' binary understandings of male/man or female/woman identities. Provider #15 fails to recognize the validity of non-binary and genderqueer identities—comparing and contrasting them to intersex people, while pathologizing the latter. As such, when a patient's goal is that of a non-binary or genderqueer presentation, the provider thinks this is effectively a request to become intersex. This conflation suggests both the psychiatric pathologization of non-binary identity (as the patient did not have a "real hormonal problem") and dismissal of non-binary or genderqueer identity ("couldn't contribute to that"). The provider therefore feels compelled to resort to psychiatric evaluation and treatment.

## Concern about unstable or undetected mental health issues

Many providers discussed the MHP evaluation as a safeguard against fears of negative consequences of initiating GAH. Providers worried that initiating GAH might exacerbate an underlying mental health condition or cause "destabilization" of a previously stable individual. This

was at times linked to an existing mental health diagnosis, but often to a more general concern, linked to providers' lack of confidence in their ability to screen for serious mental illness. In these cases, the providers assume that the MHP would support both the patient and the provider, through their clinical competencies and as an act of risk management. Regarding MHP assessment as a preventative measure, a provider stated:

> Provider #14: "I don't want to create mental health challenges for the person, so I do require a letter."

The underlying presumption here is that initiating gender-affirming hormones may "create mental health challenges," and that a letter from an MHP is an adequate way to ensure that such "challenges" are screened for and addressed. It is possible that having had experiences of caring for patients who committed suicide, a provider may search for additional ways to safeguard their patients—often turning to letters as means to do so. As the following provider shared:

> Provider #13: "When I first started, no, we didn't do a lot of that [requiring a letter, DS]. And that practice changed. . . to require mental health because I got burned a couple of times. . . I had personally two individuals that took their own lives because their mental health status was not good."

This provider had two traumatic experiences with patients. Their position is now to require letters before starting hormones—suggesting a view that mental health problems need to be addressed prior to GAH treatment despite the known benefits of GAH. If the concern is that mental health problems may be missed without the letter requirement, why not start hormones and encourage patients to see an MHP concurrently? There is an unsubstantiated sense that hormones *themselves* might cause ill effects on mental health, and that a letter is an adequate and sufficient way to ensure that such effects are addressed.

> Provider #17: "Before I do hormone therapy, I want to set up some supports for you, okay. And then, we can move ahead. . . I want to make sure your hormones don't get interrupted, that you are able to take them on time, that they are not going to cause any significant effects for you, and so that you have the support you need to make this successful and uninterrupted."

This provider justifies the need for mental health support as a way to ensure a "successful and uninterrupted" initiation of hormones. The fixation on management and control of adherence to GAH—which may well not align with patients' goals, priorities, or abilities—is an expression of a paternalistic approach (even if potentially grounded in a worry about the relatively high rates of depression and suicidality among transgender people). The above providers saw establishment of mental health care as protective against new or ongoing "challenges" including hormone interruption, and thus required a letter from a mental health provider. Furthermore, they understood GAH not as protective against worsening of mental health, but rather as a potential trigger for worsening mental health, contrary to existing literature [24, 25, 28–32].

Some providers sought to safeguard patients against the difficulties of upcoming social changes with additional mental health evaluations. They attributed this "protective" attempt to address the social and structural challenges that transgender people face in their everyday lives. One provider who universally required letters—often from specific MHPs—described an ideal

patient as having a normative life in a supportive social environment, whereas an unstable or non-supportive social environment would elicit further evaluation by a psychiatrist.

> Provider #15: "I like people to have felt that they have been this way for years, people that already introduced themselves as the other gender, people who are from a supportive family environment, someone who dress a certain way, introduce themselves a certain way for a while, there's no question of any underlying psych disorders that I would pick up based on medications; they are holding a job. You know, some of those things that have been looked at from a research point of view as just the best candidate, you know, individuals who are coming from very stable lives, those are the people I feel comfortable with."

This definition of the ideal patient excludes a large proportion of transgender people (and its "normative" elements may well exclude many cisgender people), thus requiring the majority of patients seeking GAH to undergo psychiatric evaluation. This provider's requirement for psychiatric evaluation is not necessarily out of concern for serious mental health issues that require psychiatric evaluation and care. Rather, the statement above is revealing of their understanding of mental health as it pertains to trans people, and indeed their understanding of what it means to be unquestionably transgender. As they describe the people they are "comfortable with," provider #15 reveals their discomfort with caring for transgender people who experience the all-too-common family and societal rejection and discrimination. While the research does demonstrate that transgender people with supportive environments tend to have better mental health outcomes (and hence perhaps are less in need of mental health support), there is no such evidence regarding any of the other criteria that the provider enumerates. These criteria are demonstrative of a very particular idea of transness as immutable, binary, public, long-standing, and coupled with economic stability—an idea that is not supported by research. In essence, this provider is not "comfortable" with a large proportion of transgender people [2]—in particular, non-binary or queer people—and justifies additional barriers for those who are already experiencing the effects of societal transphobia. By implying that only those with stable family and life circumstances can cope with starting GAH, this provider is missing the fact that lives tend to improve when transgender people receive the GAH they require.

## Professional expertise in diagnosis of transgender identity and mental illness

Some providers framed the MHP referral in terms of a struggle to identify the "appropriate candidate" for GAH. They claimed not to have the necessary expertise to identify such candidates and effectively outsourced this task to the MHP.

> Provider #13: "They have to have that diagnosis [of Gender Dysphoria–DS] given to them by an individual who has expertise in this space. Well, I don't like to get into the space of making a diagnosis in terms of gender dysphoria because I don't feel like I have that expertise and I don't really have that time to go through initial evaluations. So, I do lean on my mental health colleagues to make all those diagnoses and make sure their psychiatric comorbidities are under control."

As these providers deferred their decision to a third party, trust in the third party was essential. Some stated that they only trusted a limited set of MHPs. The provider above understands transness as a mental health diagnosis—understandable given the WPATH reliance on a

diagnosis of gender dysphoria per DSM criteria. There is a perhaps also understandable confusion about the potential overlaps and distinctions between transness, gender dysphoria, and the need for gender-affirming care—all of these need not necessarily align, even if current practices, including those driven by insurances, tend to equate them.

## Providers who do not require MHP evaluation

The themes identified among providers who did and did not require MHP evaluation were different. In this group of providers who did not require MHP evaluation, we identify the following themes: a) opposition to gatekeeping; b) concern about the negative implications of requiring an MHP evaluation; c) the availability and expertise of MHPs; and d) understanding of gender identity as a social construct. In this section, we describe each of these themes in greater detail. Additional advantages and disadvantages of involving an MHP, as described by providers across the board, are summarized in Fig 2.

## Opposition to gatekeeping

Several providers explicitly rejected the role of "gatekeeper." This rejection stemmed from a sympathetic and normalizing view of transness.

> Provider #9: "I think sort of requiring mental health evaluation for somebody who isn't particularly dysphoric is just contributing to the pathologizing of being trans."

The rejection of the gatekeeper role also related to providers' perceived professional role vis-à-vis the patient.

> Provider #9: "How I think about the informed consent process is that, you know, it's not my job to police somebody's lived experience. It's my job to help–a step to facilitate it and not to put up barriers as a medical professional."

This provider viewed a patient's lived experience as the ultimate determinant of their need for hormones. Furthermore, the provider here rejected the idea that they should play a part in deciding who should and shouldn't start GAH—a role of policing rather than facilitation.

Opposition to gatekeeping was linked by some providers to their view of gender as a social construct. Some providers shared their insight into the evolution of their ideas about transgender identity.

> Provider #16: "I think I have understood more, or had more of the Kool-Aid depending on how you see it, on gender as a construct and there being a spectrum. So, to say that you need to meet some bar to change your phenotype, it makes less and less sense to me."

This provider describes their own learning process, with increased understanding of gender as a continuous, social construct. Their newfound understanding, which they acknowledge others might view as ideologic following, has encouraged them to move their practice away from the WPATH model and toward IC. For providers such as #16 and #9, understanding gender as continuous, contingent, and potentially fluid implied that healthcare providers had no ability to "ascertain" transness; only the patient themselves could proclaim their need for GAH.

Others linked their rejection of "gatekeeping" directly to a dismissal of the negative consequences of GAH. Providers in the WPATH model (see previous section) raised concern for

such negative consequences as a rationale for their insistence on screening by an MHP. However, providers who rejected the need for screening addressed and dismissed this concern directly.

> Provider #10: "Frankly, we have more dangerous medications out there which don't have these rules and barriers so, from the get-go, it made no sense."

> Provider #8: "My personal philosophy is that it's pretty low risk. It's their choice as long as they are able to make an informed decision, I am happy to support them in that."

These providers compared GAH to other medical interventions favorably, making the case for reduced scrutiny. Furthermore, in the same group, many providers voiced concern over the risks associated precisely with such scrutiny, as we describe in the following section.

## The negative implications of requiring an MHP evaluation

Several providers raised concerns that restrictive approaches may result in patient self-censoring or will decrease patients' trust in their physician. One provider shared examples of how strict scrutiny of patients' gender identity can lead patients to be less truthful with their providers.

> Provider #3: "So, no trans guy or trans woman would ever say [to a healthcare provider– DS] that they wanted to be biologic parents. That's just another example of how, you know, the narrow understanding of what trans identity has removed the ability for trans people to share their truth."

This participant describes that a potential harm of strict MHP requirements is the erosion of patient trust and sharing of important psycho-social information, including parenting desires. This fear of disclosure to providers is rooted in a narrow understanding of trans identity among medical practitioners. Providers have assumed that transgender people do not desire, or worse, are not fit for, biological parenting. In some instances, trans people have been expected to forgo biological parenting by undergoing sterilization prior to initiating gender-affirming care [26]. Trans people, knowing that their expression of parenting desire may clash with traditionalist assumptions about trans identity, may conceal it. Similarly, they may conceal or refrain from sharing other important psycho-social information and shape their narrative to align with providers' expectations, thus expediting their treatment and care. Provider #3 explains that the enhanced scrutiny of trans people's narratives in the form of the required approval by an MHP is likely to increase the tendency to withhold information.

Providers who did not require a letter also discussed the positive effects of GAH, raising concern about erecting barriers to this essential care. These providers saw GAH as protective against negative mental health outcomes.

> Provider #1: "Stress, depression, mood stuff. Geez. People would come in with six different psychiatric diagnoses from various different providers over the years and. . . when they finally came down to it and realized what they were dealing with, a lot of that stuff would fall away."

> Provider #7: "I mean people are just so happy at their follow up visits after. You know, I have them come back in three months and they are just so, like, such different people when they come back. . . In my experience, I've found that starting hormones actually helps with mental health."

Provider 1 points out that rather than resolving issues, prolonged mental health care delayed much needed GAH. Furthermore, their experience was often that initiating GAH would lead to significant resolution of presumed mental health conditions. In this way, they reframe narratives about risk to one that centers the benefits of GAH. They highlight the potential downsides of requiring MHP prior to GAH, thereby delaying it.

In general, we observed that the concern for negative consequences of requiring an MHP evaluation formed a strong rationale against the requirement. Finally, the lack of access to knowledgeable and affirming MHPs played a role in this groups' tendency to skip this barrier.

## MHP availability and expertise

Among providers who did not require an MHP letter, several voiced a concern about the lack of MHPs who are knowledgeable about and respectful of diverse gender identities.

> Provider #3: "I don't ever want to have a requirement for a letter from a behavioral health person who may not have had training themselves–I mean, that really doesn't make sense to have someone who doesn't know anything about trans identity say yes, you are not trans or no, you are not trans enough–which did happen quite a bit."

> Provider #4: "It's hard to find a therapist who is a good match, no matter what. And then, it's even harder to find a therapist who is a good match and is versed in trans health and able to give respectful care to trans people."

For these providers, in addition to the mere fact of asking patients to undergo another step of evaluation prior to initiating GAH, the lack of gender-affirming MHPs created a direct barrier to care. Furthermore, they reason that the paucity of affirming and knowledgeable MHPs could place patients at risk of seeing a non-affirming or hostile MHP. These providers recognize that patient knowledge and expertise about their own identities often surpasses that of an MHP. Thus, demanding a letter from an MHP was unnecessary at best, and harmful at worst. These concerns added to the balance of reasons tipping the scale away from a letter requirement and toward an approach that decreases gatekeeping and minimizes the risks to the patient from a mandated MHP evaluation.

There is a curious similarity between the language that both groups of providers use regarding MHP expertise. Ostensibly, each group is concerned about the paucity of expertise; however, providers who require an MHP letter are led to this concern from a deeply pathologizing and/or narrow view of transness, which then requires a special set of skills to identify "true" transness. They therefore increase the scrutiny on GAH eligibility, and occasionally require letters from specific providers vetted by the provider. On the other hand, providers in the second group were primarily concerned about whether the MHPs were affirming and supportive of patients. The expertise they were seeking was not around the diagnostic process—which they saw as moot, as the patient provided that element themselves—and their concern was focused on patient access to care. Notably, the group requiring letters did not raise this latter issue—access to care—as a concern.

## Discussion

Two central questions guided providers' processes for initiating GAH. The first is that of authority—who decides on a patient's eligibility to initiate hormones. Providers relied less on MHPs when they leaned toward leaving this determination to the patient, trusting the patient's own report of their transness as sufficient. Such an approach was common among providers who saw transness as an identity, or a self-understood phenomenon, rather than as a condition

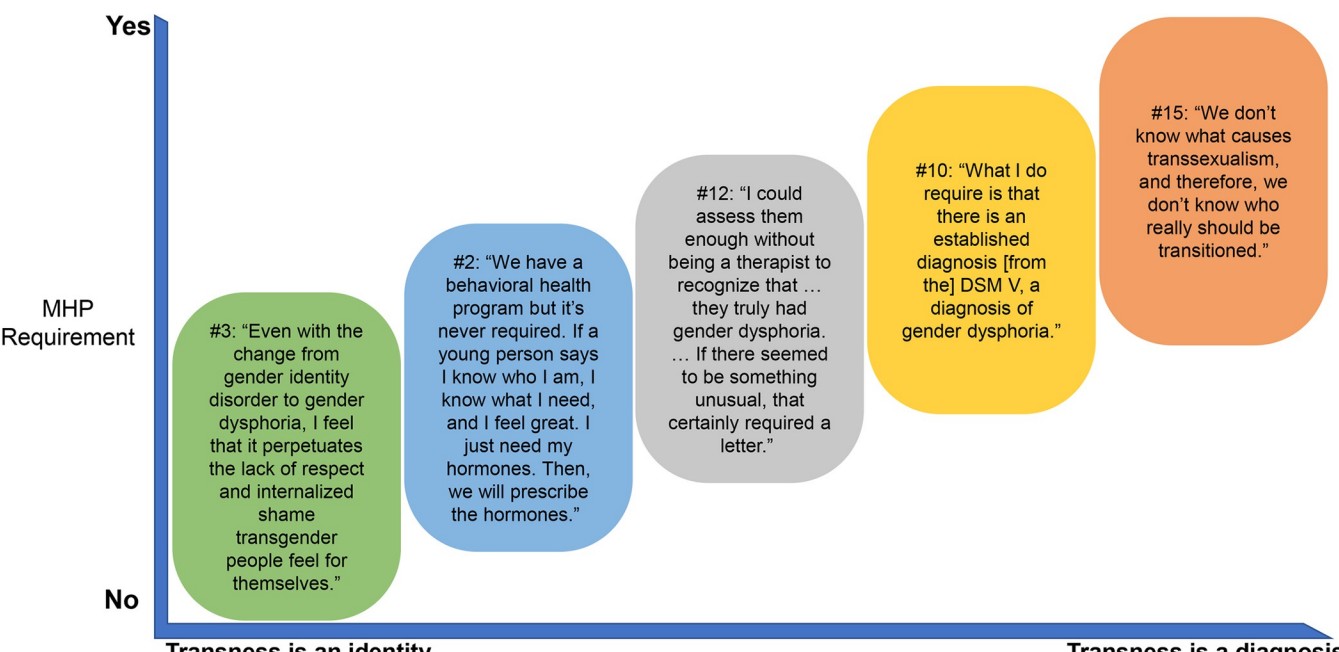

**Fig 3. Providers' attitudes about transness.** DSM, Diagnostic and Statistical Manual of Mental Disorders; MHP, mental health provider.

to be diagnosed (Fig 3). As such, the only authority on a person's gender identity is the person themselves and mediation by an MPH is not only unnecessary, but potentially detrimental to their care. On the other hand, providers who understood transness through a medicalized lens tended to seek confirmation of eligibility, outsourcing that determination to an MHP. For them, a formal diagnosis of gender dysphoria, relegated to an MHP, was seen as part of a diagnostic process prior to treatment of a (mental health) condition; the patient's self-determination was not sufficient (Fig 3). Additionally, in this group of providers, a narrow and (outdated) binary view of gender and transness was common and providers expressed misunderstanding of non-binary and genderqueer identities. Thus, the first central question differentiating the providers who follow WPATH versus those who use IC is whether the patient is to be trusted as best-positioned to report their own experience or whether the involvement of a MHP is needed.

The second question at the core of GAH initiation is that of the connection between mental illness and transness (Fig 4). While none of the providers who used the WPATH model explicitly described transness as a mental illness, the WPATH model is a medicalizing one; both the WPATH organization and the Standards of Care were developed over the past near century—primarily by MHPs, who understood transness as a "deviance" or "pathology" [27]. Providers who followed the WPATH model were persistently concerned about mental illness, "destabilization," and "serious issues" among their transgender patients with GAH initiation, occasionally based on anecdotal experience, but not supported by the literature. This is contrary to a large body of existing literature that has consistently found that both gender-affirming hormones and gender-affirming surgeries contribute to increased self-esteem, family support, and quality of life and interpersonal relationships, while reducing concerns about gender-related discrimination and violence [24, 25, 28–32]. It is well-documented that transgender people are at a vastly increased risk for anxiety/depression, suicidality, and self-injury [25, 33, 34]. However, the reasons for this are quite complex and often rest in the lack of access to gender-affirming support (including medical care and hormones) and the significantly higher rates of

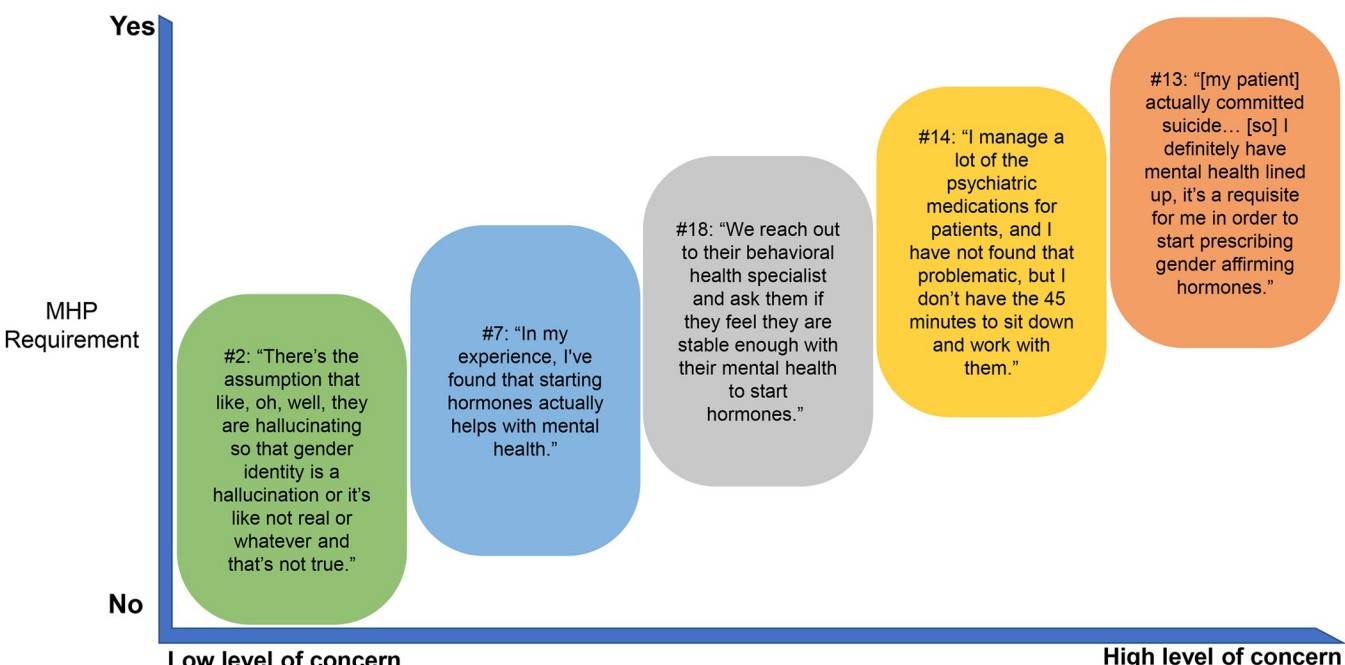

**Fig 4. Providers' concern about mental illness and effects of GAH on mental health.** GAH, gender affirming hormones; MHP, mental health provider.

harassment, victimization, and violence that transgender people experience relative to their cisgender peers [35]. In fact, gender-affirming care and gender-affirming behavior decrease risk of mental health conditions to that of general population averages [24, 25]. These providers' concerns over "destabilization" and worsening mental health lead them to view the entire initial interaction with their transgender patients through the lens of mental health. They then relegated the decision on GAH initiation to an MHP and were more likely to require a letter (Fig 4).

Providers relied less on MHPs when they saw severe mental health disorders as outliers rather than the norm among transgender people and felt comfortable screening for them. Additionally, there seemed to be a difference between the provider groups in the anticipated effects of GAH on existing mental health. In the group that used an IC model, several providers noted the possibility of improvement of mental health with hormones [24, 28, 30, 32], or were not concerned about a patient's ability to adhere to the hormones; rather, they focused on improving access to GAH and removal of any potential barriers, including letter requirements. In the group that required letters, providers voiced a concern about mental health deterioration following hormone initiation (Fig 4). Overall, providers who expressed a perception of initiating GAH as a process no different from other healthcare were more likely to use a direct approach to prescribing GAH with fewer requirements.

We posit that what drives providers' arguments for soliciting MHP evaluations and documentation is a combination of distrust in transgender patients; outdated views of transness; and a view of transness as a mental health condition in and of itself, evoking fears about serious mental illness. Their arguments thus fall into two themes: 1) The need for reassurance of the patient's stated gender identity; and 2) the need for reassurance that the patient is mentally "stable enough to transition." At the heart of this lurks an unstated doubt about transness itself, supposed difficulty in diagnosing it, and its association with mental health instability [9]. There is a perception that the stakes of "mistakes" are very high (either treating an

inappropriate candidate, or "destabilizing" someone with mental illness). Initiation of GAH was perceived by providers with a universal MHP requirement as a fragile or susceptible moment, despite the association of GAH with improved rather than worsened mental health. Providers that require an MHP letter assumed that the requirement would decrease the supposed risks of regret or destabilization—concerns that were not expressed by providers who used an IC model.

While all providers (except one) either expressed trust in the patient, or directly denied that they in any way doubted transgender people's self-account, the universal referral to an MHP by some providers both reveals and is designed to ameliorate this same doubt. These providers relegated the decision to a third party—the MHP—distancing themselves from it for reasons of expertise or time constraints. Indeed, their statements often revealed a lack of expertise or a misunderstanding of transness. Conversely, for those who do not universally require an MHP evaluation, trust in the patient, accompanied by low levels of concern for association with or worsening of mental health issues, enabled discussion of the significant disadvantages of the universal MHP requirement and a focus on a patient-centered process.

## Continuum

It is important to note that, despite the self-identification of most providers as using either "the WPATH model" or "IC," there was a wide variety of approaches to the role of MHPs in the process of initiating GAH (Fig 1). For example, among providers who require an evaluation by an MHP, some will only trust letters from known or in-house MHPs. There was a range of practices among providers who identified as using IC, from a single appointment to several appointments, with some but not all providers requiring a written consent form. At least one provider whose patients all underwent an MHP evaluation identified as using the IC model. Thus, the approach to GAH initiation is more accurately described along a continuum rather than as binary—from strict requirements for MHP evaluation to no requirement for MHP evaluation, and a spectrum of practices between the two (Fig 1).

While the discourse in the field tends to divide approaches to GAH into either "IC" or "WPATH model," our interviews reveal complex and continuous solutions that providers develop individually to manage their relationships with transgender patients. Despite the appearance of professional consensus in the form of the WPATH Standards of Care (or Endocrine Society guidelines, which are very similar in this regard), many providers identify gaps in or inadequacies of these guidelines to address clinical realities, with significant concern raised about the need for—and harm from—the recommendation for universal approval by MHPs. Instead, these providers develop practice patterns that suit their perceptions and practical needs, often depending in large part on the availability of MHPs within their professional networks or leaving the navigation of the guideline inadequacies and structural barriers to care to their patients.

While we identified no consistent trends in terms of provider approach and the path they took to becoming a GAH provider, we did hear from providers (regardless of approach to MHPs) that they encountered their first transgender patient at a late stage in their career. It is clear that medical education on transgender care needs to be improved and included at all levels, from medical school to postgraduate training. Such training must also confront transphobia directly [36] and engage transgender people as expert educators.

## Limitations

This study was limited by some lack of diversity among respondents, with a majority white and cisgender sample. While as a qualitative study, our results do not purport to be

"generalizable" nor our sample representative, we do acknowledge that this lack of racial diversity is a significant limitation. It may be linked to a paucity of Black and other non-White providers throughout medicine in general, and in specific circles in medicine in particular. Furthermore, while the WPATH guidelines are international, our study was limited to providers in the United States and may not reflect practice patterns in other countries. Though a minority of providers identified as using the WPATH model, greater de-facto diversity in practice patterns emerged from interviews and we were able to reach thematic saturation.

## Conclusions

Amongst the providers we interviewed, the assumptions about transness underpinning a universal requirement for mental health evaluation are rooted in perceptions of transness as mental illness. These assumptions include relegation of the determination of "true" transness from patients to MHPs and the unique and particular concern about mental illness among transgender people. Many of our interviewees described harmful effects of these outdated assumptions. Such effects must be acknowledged and addressed in unified guidelines.

An alternative understanding of transness—not as a condition to be diagnosed by MHPs, but as an identity—was suggested by some participants. Such an approach enables removal of the MHP letter barrier, increasing access to care. The vast majority of our participants did not raise concern for increased risk to patients with removal of this barrier. Future research into perspectives of transgender people, as well as those of MHPs who care for transgender patients, will be useful along with our data in creating an evidence base for clinical guidelines and recommendations. Quantitative evaluation of providers' approaches may further elucidate trends in the field.

Based on our interviews, the division between providers who either require or do not require an MHP letter is a false dichotomy, as there is a range of practices even within our small sample, and providers use the terms "informed consent" and "WPATH model" to refer to a wide range of practices. This range should be acknowledged and made transparent in order to enable patients to have more accurate expectations as they start GAH, and should be addressed in future guidelines.

This study revealed the transphobia underlying requirements for universal MHP referral, and provides evidence from prescribers who witnessed the downsides and risks of the current guidelines. Given this evidence, and the known benefits of GAH, future guidelines should focus on increasing patient access to care through removal of such requirements. This will enable greater provider flexibility and build in a patient-centered, normalized, and destigmatizing approach to prescription of GAH. While we recognize the mental health disparities experienced by trans people, which are mediated by societal factors, and support increasing access to mental health services for those who want it, the concern for transgender people's mental health—at a population level or among particular individuals—is an inadequate justification to require a MHP evaluation for all trans people seeking GAH.

## Acknowledgments

The authors wish to thank Jesse Ballard, Elliot Popoff, and Racquelle Trammell for their support of this project, their invaluable feedback, and their participation in the Trans Health Advisory Board for the community member arm of this study.

## Author Contributions

**Conceptualization:** Daphna Stroumsa, Molly B. Moravek, Justine P. Wu.

**Data curation:** Leah A. Minadeo, Mariam Maksutova.

**Formal analysis:** Daphna Stroumsa, Rob Stephenson, Paul N. Pfeiffer, Justine P. Wu.

**Funding acquisition:** Daphna Stroumsa, Justine P. Wu.

**Investigation:** Daphna Stroumsa, Leah A. Minadeo, Mariam Maksutova, Justine P. Wu.

**Methodology:** Daphna Stroumsa, Justine P. Wu.

**Project administration:** Leah A. Minadeo.

**Supervision:** Justine P. Wu.

**Validation:** Leah A. Minadeo.

**Visualization:** Mariam Maksutova.

**Writing – original draft:** Daphna Stroumsa, Justine P. Wu.

**Writing – review & editing:** Daphna Stroumsa, Leah A. Minadeo, Molly B. Moravek, Rob Stephenson, Paul N. Pfeiffer, Justine P. Wu.

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
