## [Decision Letter · Decision Letter 0]

13 Apr 2021

PONE-D-20-30031

It’s not binary: a qualitative study of providers’ approaches to gender-affirming hormone initiation and the two-model fallacy

PLOS ONE

Dear Dr. Stroumsa,

Thank you for submitting your manuscript to PLOS ONE. After careful consideration, we feel that it has merit but does not fully meet PLOS ONE’s publication criteria as it currently stands. Therefore, we invite you to submit a revised version of the manuscript that addresses the points raised during the review process.

The manuscript has been evaluated by two reviewers, and their comments are available below. You will see the reviewers have commented on the relevance of your work. However, they have also raised a number of concerns that should be addressed before the manuscript can be further considered for publication.

The key concern noted by Reviewer 1 relates to the analysis and reporting of the data. Specifically, Reviewer 1 requested clarity regarding the analytic approach and additional interpretation of the data points. These issues impact the overall conclusions of the manuscript and should be explored.

We look forward to receiving your revised manuscript.

Kind regards,

Danielle Poole

Staff Editor

PLOS ONE

Journal Requirements:

Reviewers' comments:

Reviewer's Responses to Questions

**Comments to the Author**

1. Is the manuscript technically sound, and do the data support the conclusions?

Reviewer #1: Partly

Reviewer #2: Yes

2. Has the statistical analysis been performed appropriately and rigorously? 

Reviewer #1: N/A

Reviewer #2: Yes

3. Have the authors made all data underlying the findings in their manuscript fully available?

Reviewer #1: No

Reviewer #2: Yes

4. Is the manuscript presented in an intelligible fashion and written in standard English?

Reviewer #1: No

Reviewer #2: Yes

5. Review Comments to the Author

Reviewer #1: I want to thank the authors for their work on this topic which is very relevant to both patients and providers vis-à-vis gender-affirming care. I appreciate the intent of the authors' research question here: "In this study, we aimed to describe the range of experiences and attitudes regarding hormone initiation among GAH providers and to assess factors that affect these attitudes, including the perceived advantages and disadvantages of each." In the abstract the aim is stated purely as descriptive, which—after reading the manuscript-- seems more accurate, less compelling or useful, and maybe even harmful.

The paper is largely descriptive and lacks a rigorous assessment piece without which I do not think the paper is publishable. I’m not sure if that is because the team is afraid of being critical of treatment approaches used by providers within their own social networks, or if it is fear of professional consequences for highlighting the lack of competence that comes with the gatekeeping model, or if the team really thinks these two models should be treated as equally legitimate approaches. Part of the problem might be that the analysis seems unfinished – I think you have some excellent quotes, organized into meaningful thematic sections, but the summaries of the data are lacking. Most of your paragraphs are largely quotes preceded by one sentence that summarizes them. I was wanting more from the authors in the results section in terms of moving beyond the quotes with synthesis and reconceptualization.

Consequently, the main finding -- that providers operate (or even "identify") on a continuum across these two “nonbinary” models -- is not terribly compelling. Models are just that – models/guideposts-- and not meant to be followed to the letter; they are meant to be adapted to contexts. So, the conclusion here that whispers at recognizing an unproblematic taxonomy of treatment approaches does not rigorously interpret the data that was provided. Maybe more importantly, the “treatment is not binary” feels a bit lazy and also degrades the idea of *gender* as something that is nonbinary and varies across time and space for individuals but also for societies, etc.

What might be helpful in the Discussion is to situate your findings within the literature on and the historical contexts of these models—where did they come from and why or in response to what? The idea presented in the intro (p. 4 line 75-76) that treatment approach is simply a matter of individual preference is not true and ignores not only institutional policies/mission&values but also each provider’s orientation to health justice/bodily autonomy, medical malpractice, social determinants of health, and the DSM and mental health, etc. What do your interviews say about what sustains these models or is informing their change/demise?

I also appreciate that in a couple places you gesture at what might be the key difference between these models: is it possible to “diagnose” someone’s gender in terms of exogenous hormone uptake? WPATH model says yes and it should be a MHP (there’s a history there that needs addressing), while informed consent says no – only the patient can really know what they might need. Just like some cis men ask for testosterone for a variety of reasons while some cis men choose other pathways for treatment, not all trans men want testosterone and some nonbinary AFAB persons also do. Cis men do not need a MHP to validate their request. Same with cisgender-affirming surgeries like facelifts and tummy tucks and a range of implants and other plastic surgeries.

Here are some more specific comments on the manuscript:

Results:

P. 10 Lines 165-170 Your paragraph on provider decision-making is a very good example of the problem with the results section. First, I cannot tell what “decision-making” (165) even refers to. Is it whether to send them to an MHP, whether to treat them at all, whether to prescribe hormones? And “decision-making for a variety of factors” is incredibly vague, providing no framework for the reader. Also, you should avoid requiring the reader to read several quotes in the mid-sentence bracketed by parentheses – it is really hard to follow. In line 170 you need to specify that you are talking about *hormonal* transition – or if you are talking about some other transition, say so, and say why these providers are concerned with that. This paragraph provides evidence of a wide range of responses but no synthesis that follows or reconceptualization. What does it mean that a provider bases their prescribing on whether a patient has insurance while another cares about whether or not they are out? These are really big statements that are left behind with no scaffolding from the author team.

173-175

I don’t think a paragraph can consist of only one sentence, a problem that appears in other parts of this paper. (This sentence is poorly written too – “the need” appears twice.) Can you say more about what this section is doing? Give the reader more of a framework from which to interpret the section. Please eliminate/build out all one-sentence paragraphs.

178-79

This sentence seems to contradict itself: “While some who required a mental health evaluation acknowledged that they ultimately rely on a patient’s self-identification as transgender.” How is it possible to claim that they really rely on the patient’s self-id when they in fact are relying on an MHP as a rule? You need to elaborate on what they were saying – maybe that they want to ensure that the patient is resilient enough to endure the social upheaval that might emerge? -- or make note of this glaring contradiction. When the author team does not either explain the quote further and/or make note of these contradictions, it is almost like a defense of this model or approach coming through. Similarly, when providers are saying for example: 1) they do not understand nonbinary identities and so they require an MHP, or 2) they got burned when two of their patients killed themselves and so they send all patients to an MHP, the author team should further elaborate on what these explanations suggest about the model they are using and about their approach/view of patients whom they are treating.

p.14, 267

Section on MHP availability and expertise is poorly organized and/or written. The paragraph leads with a very important observation, but the rest of the first paragraph is one quote. My rule of thumb is to *never* end a paragraph with a quote, whereas most paragraphs in your Results section end with a quote and simply move on to the next theme. The data needs restating/summarizing/linking/synthesis that nods at your broader argument or toward what is coming in the next section, etc. The Results is very choppy as currently written.

p. 15, 314-333

Same concerns here as preceding comment. This section is trying to do way too much without providing clear structure, clear writing, and clear scaffolding/framing for the reader to understand what to do with the data. It consists of 5 quotes and 2.5 sentences of your own – a “healthy” paragraph might have four sentences and one or two quotes.

p. 17, 350-353

Provider #17 appears in the section about strict requirement of MHP (p.11, line 181). In this paragraph they are referred to as a provider who does NOT universally send to an MHP.

There are lots of rich data points in this manuscript, but I think the team needs to figure out what is the most compelling finding and completely rewrite the Discussion section. What can you say about these models and the interests they serve? I encourage the team to comb through each paragraph in the Results and rewrite it with stronger frameworks and syntheses that generate a more coherent critique and move the literature forward. A section that felt absent, that may help you critique the models more rigorously, is some kind of synopsis about how these providers came to care for trans/nonbinary patients (i.e. one of your interview domains to explore). What was it about their training? What was lacking or what informed it that brought them to these different approaches? I will say, for example, that very well-meaning providers/colleagues, who use the WPATH model, still think that using the “genderbread man” or the “gender unicorn” is a useful pedagogical tool to teach family medicine residents about gender-affirming care. What does that say about the level of medical sophistication around gender and gender-affirming care? I think the lack of non-White – and specifically Black providers – also limits your takeaways. I don’t know any Black providers who are members of WPATH or who use the WPATH model. Why do you think that is? Well, you cannot say because you don’t have the data, but you might check the literature on this/the history of racism in medicine and mental healthcare/the high cost of WPATH membership/the fact that many Black providers who do trans care do so at safety net clinics and health centers.

There are many places where citations are missing. This paper cites literature very sparsely. There is a rich literature on these models and on the associations between GAH uptake and mental health benefits, but I do not see much of them cited. Similarly, there is literature on reasons why patients might not disclose gender identities fully to their providers, particularly nonbinary patients, but these studies don’t seem to be cited or used to advance an argument/critique either. Another rich literature to draw upon is that of medical education and training (or lack thereof or inadequacy of). I think this paper can be very rich and make a valuable contribution, but in its current form, it does not.

Reviewer #2: This is an extremely well written and important manuscript. These data are nuanced and complex and will add a missing component to the dearth of research on provider practice and motivation around gender-affirming hormone care. Well done.

6. PLOS authors have the option to publish the peer review history of their article (what does this mean?). If published, this will include your full peer review and any attached files.

Reviewer #1: No

Reviewer #2: No

---

## [Author Response · Author response to Decision Letter 0]

9 Sep 2021

Comments to the Author

1. Is the manuscript technically sound, and do the data support the conclusions?

Reviewer #1: Partly

Reviewer #2: Yes

2. Has the statistical analysis been performed appropriately and rigorously?

Reviewer #1: N/A

Reviewer #2: Yes

 3. Have the authors made all data underlying the findings in their manuscript fully available?

Reviewer #1: No

Reviewer #2: Yes

4. Is the manuscript presented in an intelligible fashion and written in standard English.

Reviewer #1: No

Reviewer #2: Yes

5. Review Comments to the Author

Reviewer #1: I want to thank the authors for their work on this topic which is very relevant to both patients and providers vis-à-vis gender-affirming care. I appreciate the intent of the authors' research question here: "In this study, we aimed to describe the range of experiences and attitudes regarding hormone initiation among GAH providers and to assess factors that affect these attitudes, including the perceived advantages and disadvantages of each." In the abstract the aim is stated purely as descriptive, which—after reading the manuscript-- seems more accurate, less compelling or useful, and maybe even harmful.

The paper is largely descriptive and lacks a rigorous assessment piece without which I do not think the paper is publishable. I’m not sure if that is because the team is afraid of being critical of treatment approaches used by providers within their own social networks, or if it is fear of professional consequences for highlighting the lack of competence that comes with the gatekeeping model, or if the team really thinks these two models should be treated as equally legitimate approaches. Part of the problem might be that the analysis seems unfinished – I think you have some excellent quotes, organized into meaningful thematic sections, but the summaries of the data are lacking. Most of your paragraphs are largely quotes preceded by one sentence that summarizes them. I was wanting more from the authors in the results section in terms of moving beyond the quotes with synthesis and reconceptualization.

Consequently, the main finding -- that providers operate (or even "identify") on a continuum across these two “nonbinary” models -- is not terribly compelling. Models are just that – models/guideposts-- and not meant to be followed to the letter; they are meant to be adapted to contexts. So, the conclusion here that whispers at recognizing an unproblematic taxonomy of treatment approaches does not rigorously interpret the data that was provided. Maybe more importantly, the “treatment is not binary” feels a bit lazy and also degrades the idea of *gender* as something that is nonbinary and varies across time and space for individuals but also for societies, etc.

What might be helpful in the Discussion is to situate your findings within the literature on and the historical contexts of these models—where did they come from and why or in response to what? The idea presented in the intro (p. 4 line 75-76) that treatment approach is simply a matter of individual preference is not true and ignores not only institutional policies/mission&values but also each provider’s orientation to health justice/bodily autonomy, medical malpractice, social determinants of health, and the DSM and mental health, etc. What do your interviews say about what sustains these models or is informing their change/demise?

I also appreciate that in a couple places you gesture at what might be the key difference between these models: is it possible to “diagnose” someone’s gender in terms of exogenous hormone uptake? WPATH model says yes and it should be a MHP (there’s a history there that needs addressing), while informed consent says no – only the patient can really know what they might need. Just like some cis men ask for testosterone for a variety of reasons while some cis men choose other pathways for treatment, not all trans men want testosterone and some nonbinary AFAB persons also do. Cis men do not need a MHP to validate their request. Same with cisgender-affirming surgeries like facelifts and tummy tucks and a range of implants and other plastic surgeries.

Author Response: Many thanks to the reviewers for the engaged, thoughtful, and helpful comments. We have significantly revised the paper and have addressed their comments to the best of our ability. We agree that the more compelling of our findings is not the fact of continuum of care, but rather the factors affecting provider’s decisions on MHP involvement. We have reframed our paper and the discussion to center more explicitly on the differences in perception of transness. We highlight some of the extensive changes below.

“Two central questions guided providers’ processes for initiating GAH. The first is that of authority—who decides on a patient’s eligibility to initiate hormones. Providers relied less on MHPs when they leaned toward leaving this determination to the patient, trusting the patient’s own report of their transness as sufficient. Such an approach was common among providers who saw transness as an identity, or a self-understood phenomenon, rather than as a condition to be diagnosed (Fig 3). As such, the only authority on a person’s gender identity is the person themselves and mediation by an MPH is not only unnecessary, but potentially detrimental to their care. On the other hand, providers who understood transness through a medicalized lens tended to seek confirmation of eligibility, outsourcing that determination to an MHP. For them, a formal diagnosis of gender dysphoria, relegated to an MHP, was seen as part of a diagnostic process prior to treatment of a (mental health) condition; the patient’s self-determination was not sufficient (Fig 3). Additionally, in this group of providers, a narrow and (outdated) binary view of gender and transness was common and providers expressed misunderstanding of non-binary and genderqueer identities. Thus, the first central question differentiating the providers who follow WPATH versus those who use IC is whether the patient is to be trusted as best-positioned to report their own experience or whether the involvement of a MHP is needed.” (Discussion Paragraph 1, Page 24)

“The second question at the core of GAH initiation is that of the connection between mental illness and transness (Fig 4). While none of the providers who used the WPATH model explicitly described transness as a mental illness, the WPATH model is a medicalizing one; both the WPATH organization and the Standards of Care were developed over the past near century—primarily by MHPs, who understood transness as a “deviance” or “pathology” [35]. Providers who followed the WPATH model were persistently concerned about mental illness, “destabilization,” and “serious issues” among their transgender patients with GAH initiation, occasionally based on anecdotal experience, but not supported by the literature. These concerns lead them to view the entire initial interaction with their transgender patients through the lens of mental health. They then relegated the decision on GAH initiation to an MHP and were more likely to require a letter (Fig 4).” (Page 25, Paragraph 1)

“Providers relied less on MHPs when they saw severe mental health disorders as outliers rather than the norm among transgender people and felt comfortable screening for them. Additionally, there seemed to be a difference between the provider groups in the anticipated effects of GAH on existing mental health. In the group that used an IC model, several providers noted the possibility of improvement of mental health with hormones [24,26,28,29], or were not concerned about a patient’s ability to adhere to the hormones; rather, they focused on improving access to GAH and removal of any potential barriers, including letter requirements. In the group that required letters, providers voiced a concern about mental health deterioration following hormone initiation (Fig 4). Overall, providers who expressed a perception of initiating GAH as a process no different from other healthcare were more likely to use a direct approach to prescribing GAH with fewer requirements.” (Pages 25-26, Paragraph 2)

Here are some more specific comments on the manuscript:

Results:

P. 10 Lines 165-170 Your paragraph on provider decision-making is a very good example of the problem with the results section. First, I cannot tell what “decision-making” (165) even refers to. Is it whether to send them to an MHP, whether to treat them at all, whether to prescribe hormones? And “decision-making for a variety of factors” is incredibly vague, providing no framework for the reader. Also, you should avoid requiring the reader to read several quotes in the mid-sentence bracketed by parentheses – it is really hard to follow. In line 170 you need to specify that you are talking about *hormonal* transition – or if you are talking about some other transition, say so, and say why these providers are concerned with that. This paragraph provides evidence of a wide range of responses but no synthesis that follows or reconceptualization. What does it mean that a provider bases their prescribing on whether a patient has insurance while another cares about whether or not they are out? These are really big statements that are left behind with no scaffolding from the author team.

Author Response: Thank you for helping us clarify this introductory paragraph. We have made the below changes to the paragraph and attempted to clarify that further analysis is to follow. 

“Providers attributed their decision about MHP involvement to a variety of factors. Themes ranged from perceptions regarding practical concerns with individual patients, to broader attitudes regarding gender identity (“Are they insured or not? Whose insurance are they on? Are they out? … I think the biggest barrier is the financial piece and the safety and security”); and broader attitudes regarding gender identity (“Some people do have gender dysphoria but not everybody is dysphoric about just being who they are”; “I think we are sometimes too rigid about how we define gender”). All providers considered the social challenges related to both social and medical transition. They acknowledged the potential social, emotional, and financial implications of potential rejections and importance of social networks in supporting a trans person as they transitioned. These themes were present in our interviews with providers regardless of their approach to hormone initiation. However, there were some distinct differences by approach. In the following sections we review the main themes identified and analyze them according to provider approach to MHP involvement.” (Page 12, Paragraph 1)

173-175

I don’t think a paragraph can consist of only one sentence, a problem that appears in other parts of this paper. (This sentence is poorly written too – “the need” appears twice.) Can you say more about what this section is doing? Give the reader more of a framework from which to interpret the section. Please eliminate/build out all one-sentence paragraphs.

Author Response: Thank you for helping us improve the clarity of our paper. We have rephrased this paragraph as follows:

“In the following section we present the themes among providers who strictly required an MHP evaluation prior to GAH initiation. Within this group, we identify the following themes: a) the importance of MHP consultation to ascertain a diagnosis of gender dysphoria; b) concern about unstable or undetected mental health issues; and c) the provider’s own confidence in addressing these issues. We detail each of these themes below. These and other advantages and disadvantages of involving an MHP are summarized in Fig 2.” (Pages 12-13, Paragraph 2)

Furthermore, we have eliminated one sentence paragraphs through the remainder of the paper. 

178-79

This sentence seems to contradict itself: “While some who required a mental health evaluation acknowledged that they ultimately rely on a patient’s self-identification as transgender.” How is it possible to claim that they really rely on the patient’s self-id when they in fact are relying on an MHP as a rule? You need to elaborate on what they were saying – maybe that they want to ensure that the patient is resilient enough to endure the social upheaval that might emerge? -- or make note of this glaring contradiction. When the author team does not either explain the quote further and/or make note of these contradictions, it is almost like a defense of this model or approach coming through. Similarly, when providers are saying for example: 1) they do not understand nonbinary identities and so they require an MHP, or 2) they got burned when two of their patients killed themselves and so they send all patients to an MHP, the author team should further elaborate on what these explanations suggest about the model they are using and about their approach/view of patients whom they are treating.

Author Response: We have added the following explanations to these particular instances, as follows:

“This provider acknowledges that in principle, they agree that transness is best attested by the person themselves. Yet, there is an apparent discrepancy between that claim and the provider’s insistence on having that identity verified by an MHP. Most physicians will not explicitly dismiss a patient’s gender identity, likely because of social desirability bias. However, ultimately, it is disbelief or doubt in the patient’s transness that determines the course of action.” (Pages 13-14, Paragraph 3)

Regarding nonbinary identities, we have added the following text: 

“Both providers (#14 and #15) display fundamental misunderstandings of non-binary and genderqueer identities or of ways of being trans that didn’t match up with their expectations about sex/gender. Providers were confused when patients with such identities did not fit the providers’ binary understandings of male/man or female/woman identities. Provider #15 fails to recognize the validity of non-binary and genderqueer identities—comparing and contrasting them to intersex people, while pathologizing the latter. As such, when a patient’s goal is that of a non-binary or genderqueer presentation, the provider thinks this is effectively a request to become intersex. This conflation suggests both the psychiatric pathologization of non-binary identity (as the patient did not have a “real hormonal problem”) and dismissal of non-binary or genderqueer identity (“couldn’t contribute to that”). The provider therefore feels compelled to resort to psychiatric evaluation and treatment.” (Pages 14-15, 3)

Following the quote from the physician who reports being burned by suicidal patients, we further explain: 

“This provider had two traumatic experiences with patients. Their position is now to require letters before starting hormones—suggesting a view that mental health problems need to be addressed prior to GAH treatment despite the known benefits of GAH. If the concern is that mental health problems may be missed without the letter requirement, why not start hormones and encourage patients to see an MHP concurrently? There is an unsubstantiated sense that hormones themselves might cause ill effects on mental health, and that a letter is an adequate and sufficient way to ensure that such effects are addressed. (Page 16, Paragraph 3)

p.14, 267

Section on MHP availability and expertise is poorly organized and/or written. The paragraph leads with a very important observation, but the rest of the first paragraph is one quote. My rule of thumb is to *never* end a paragraph with a quote, whereas most paragraphs in your Results section end with a quote and simply move on to the next theme. The data needs restating/summarizing/linking/synthesis that nods at your broader argument or toward what is coming in the next section, etc. The Results is very choppy as currently written.

Author Response: We have restructured the results section, including reordering of several sub-sections, and ensured summary and further analysis following quotes. We hope that this makes for a smoother reading experience and interesting analysis. 

p. 15, 314-333

Same concerns here as preceding comment. This section is trying to do way too much without providing clear structure, clear writing, and clear scaffolding/framing for the reader to understand what to do with the data. It consists of 5 quotes and 2.5 sentences of your own – a “healthy” paragraph might have four sentences and one or two quotes.

Author Response: We have added significant summary and analysis for this section as well. 

p. 17, 350-353

Provider #17 appears in the section about strict requirement of MHP (p.11, line 181). In this paragraph they are referred to as a provider who does NOT universally send to an MHP.

Author Response: We have addressed this and changed this paragraph. 

There are lots of rich data points in this manuscript, but I think the team needs to figure out what is the most compelling finding and completely rewrite the Discussion section. What can you say about these models and the interests they serve? I encourage the team to comb through each paragraph in the Results and rewrite it with stronger frameworks and syntheses that generate a more coherent critique and move the literature forward. A section that felt absent, that may help you critique the models more rigorously, is some kind of synopsis about how these providers came to care for trans/nonbinary patients (i.e. one of your interview domains to explore). What was it about their training? What was lacking or what informed it that brought them to these different approaches? I will say, for example, that very well-meaning providers/colleagues, who use the WPATH model, still think that using the “genderbread man” or the “gender unicorn” is a useful pedagogical tool to teach family medicine residents about gender-affirming care. What does that say about the level of medical sophistication around gender and gender-affirming care? I think the lack of non-White – and specifically Black providers – also limits your takeaways. I don’t know any Black providers who are members of WPATH or who use the WPATH model. Why do you think that is? Well, you cannot say because you don’t have the data, but you might check the literature on this/the history of racism in medicine and mental healthcare/the high cost of WPATH membership/the fact that many Black providers who do trans care do so at safety net clinics and health centers.

Author Response: We are very grateful to the reviewer for these insightful comments. We hope that we have managed to address them by significantly restructuring and rewriting both the results and the discussion sections.

While our interviews contained data about the paths providers took to become involved in hormone prescription, our findings do not allow us to make broad categorizations or come to significant conclusions regarding patterns in training and their effect on provider approach to hormone prescription.

We could not agree with you more regarding both the lack of diversity in medicine, and the “flatness” and lack of nuance in medical training - not solely regarding gender and sexuality, but more broadly as it relates to many social questions, including racism in medicine and in society at large.

There are many places where citations are missing. This paper cites literature very sparsely. There is a rich literature on these models and on the associations between GAH uptake and mental health benefits, but I do not see much of them cited. Similarly, there is literature on reasons why patients might not disclose gender identities fully to their providers, particularly nonbinary patients, but these studies don’t seem to be cited or used to advance an argument/critique either. Another rich literature to draw upon is that of medical education and training (or lack thereof or inadequacy of). I think this paper can be very rich and make a valuable contribution, but in its current form, it does not.

Author Response: Thank you for drawing our attention to this. We have thickened both our analyses and our citations and aimed to base our argumentation on existing literature. 

Reviewer #2: This is an extremely well written and important manuscript. These data are nuanced and complex and will add a missing component to the dearth of research on provider practice and motivation around gender-affirming hormone care. Well done.

6. PLOS authors have the option to publish the peer review history of their article (what does this mean?). If published, this will include your full peer review and any attached files.

Do you want your identity to be public for this peer review? For information about this choice, including consent withdrawal, please see our Privacy Policy.

Reviewer #1: No

Reviewer #2: No

---

## [Decision Letter · Decision Letter 1]

17 Feb 2022

PONE-D-20-30031R1

Initiating gender-affirming hormones for transgender and non-binary people: A qualitative study of providers’ perspectives on requiring mental health evaluations

PLOS ONE

Dear Dr. Stroumsa,

Thank you for submitting your manuscript to PLOS ONE. After careful consideration, we feel that it has merit but does not fully meet PLOS ONE’s publication criteria as it currently stands. Therefore, we invite you to submit a revised version of the manuscript that addresses the points raised during the review process.

We apologize for the delays you have experienced in the review process. The manuscript has been evaluated by three reviewers, and their comments are available below. As the previous Academic Editor has become unavailable, we have additionally consulted with the Section Editor overseeing this area of the journal, who has provided the comments below.  Could you please revise the manuscript to carefully address the concerns raised?

We look forward to receiving your revised manuscript.

Kind regards,

Vanessa Carels

Staff Editor

PLOS ONE

Journal Requirements:

Comments from Section Editor:

 First, it must be said that the qualitative methodology is very well used in terms of description of the procedures and the reporting of the results.

My concern is with the conclusions. I feel that the first two paragraphs of the conclusion should be followed by the last paragraph of the Continuum section. The last 3 paragraphs of the conclusion slightly extrapolate the objectives and results of the study and seem to me to make strong statements about possible consequences for the collective and individual health of trans people, but which go beyond the present article, which is exploratory in nature.

These paragraphs and also the conclusion of the abstract are categorical as if they came from a research of a descriptive/quantitative nature and deserve more nuance.

Furthermore, I feel that the discussion and conclusion should focus on the US context as this issue is more acute in the context in which the research was situated and is present in a different way in other parts of the globe.

Reviewers' comments:

Reviewer's Responses to Questions

**Comments to the Author**

1. If the authors have adequately addressed your comments raised in a previous round of review and you feel that this manuscript is now acceptable for publication, you may indicate that here to bypass the “Comments to the Author” section, enter your conflict of interest statement in the “Confidential to Editor” section, and submit your "Accept" recommendation.

Reviewer #1: (No Response)

Reviewer #2: All comments have been addressed

Reviewer #3: All comments have been addressed

2. Is the manuscript technically sound, and do the data support the conclusions?

Reviewer #1: Partly

Reviewer #2: Yes

Reviewer #3: Yes

3. Has the statistical analysis been performed appropriately and rigorously? 

Reviewer #1: N/A

Reviewer #2: Yes

Reviewer #3: Yes

4. Have the authors made all data underlying the findings in their manuscript fully available?

Reviewer #1: No

Reviewer #2: Yes

Reviewer #3: Yes

5. Is the manuscript presented in an intelligible fashion and written in standard English?

Reviewer #1: Yes

Reviewer #2: Yes

Reviewer #3: No

6. Review Comments to the Author

Reviewer #1: Thank you for the opportunity to read the revised version of the manuscript. It is a much stronger, coherent, and well-written paper. There are minor, albeit fairly fundamental, revisions I would suggest making.

1) There remains conceptual murkiness around the fundamental concept discussed in this paper - gender identity - or maybe just identity itself. First, the last sentence of the intro where "transgender" is defined is not useful at all. It's circular and also just incorrect. What are non-cisgender identities? For example, trans AND cis adults alike identify as men or as women. Please provide a more nuanced and careful definition.

Relatedly, on the next page, in methods, some clarifications may be helpful. I'm not sure what the reflexivity section is trying to do. The second sentence is very confusing - either say your team consists of "cisgender or non-binary as well as straight and queer" individuals or say it consists of "a diversity of genders and sexualities". As it currently stands I'm not sure what you are trying to say about cisgender, queer, and non-binary people on your team that is unique from the diversity of sexualities and genders. Is queer a gender? Are non-binary people asexual? Are cisgender people straight? Why have those three been singled out from the diversity? Further there seems to be a sentence missing about why it's important to note that no one on the author team has ANY experience with seeking medical transition and how that may impact the findings in this paper. Stating your privilege as researchers would remain true had you included a co-author with this lived experience.

In the next paragraph, purposive sampling was used to recruit providers who "self-identify" as using WPATH or IC. That term is really confusing first of all. To my knowledge, providers do not conflate their treatment model preferences with self-identification. Secondly, it doesn't seem like that inclusion criteria is true. Table 1 says that 17% of your sample uses "other" as their treatment model/guidance. Please align your methods and inclusion criteria with your results.

2) Page 12/line 188-190 there is a redundant theme stated about "broader attitudes regarding gender identity" with very different quotes as examples that follow this same theme - please clarify what is different about these same themes by way of these quotes.

3) Transsexual is spelled differently in different parts of the paper. Technically, it has two "s"s but some adhere to "transexual" (i.e. not unlike nonbinary vs. non-binary). Please choose which version you are meaning to use and stick with it.

4) The quote from #16/line 391 should be modified or commented upon. Why did you think it is important to include a quote, referred to as "their insight," where an ostensible ally is saying that many seem to think this provider has "had more of the Kool-Aid" because they do not gatekeep? Please interpret or share the importance with the reader or don't include it. As it stands, inclusion of this analogy perpetuates the idea of a growing trans medical industrial complex as driven by a cult like that of the Jim Jones kind where it ends in mass suicide for mostly black Americans. Do you think the provider's ideas have "evolved" as you say prior to the quote? I think they sound ambivalent/conflicted about their choices and/or unfairly persecuted by acting as an ally. Maybe say more about what is going on here; otherwise I would not include the "Kool-Aid" comment as an insight around an evolution of ideas.

5) Granted i have not had time to do as careful a read of the discussion as I'd like, it seems much stronger and coherent and comprehensive than the initial version (the "Continuum" subheader may need something more, and you might add a "Limitations" subheader which currently falls under "Continuum". Thank you so much for the thoughtful revision.

Reviewer #2: (No Response)

Reviewer #3: This is an important topic, and I appreciate what your qualitative study is able to add to the discussion around MHP letter requirements before initiation of gender-affirming care.

I said "no" for number 5 because some of the additions made after the first round of reviews disrupted the flow of the manuscript.

While commenting on and contextualizing the quotes is important, the placement of your critique immediately after the quotes in the results (for example, in lines 229-232) made me feel the opinion of the authors in a way that was distracting as a reader - even though I agree with your interpretation/critique/additions.

To back up your critiques of the quotes of the respondents who required letters, you included many more citations in this draft, which was important. Yet again, I think that discussing the quote within the context of available literature (for example, lines 275-285) would have been more appropriate in the discussion section. In fact, I found myself wondering who providers 13-15 were, and getting upset that they feel qualified to do gender-affirming care. Perhaps this was the intent, but it made it difficult for me to maintain attention through the rest of the manuscript. And, as someone who has been quoted out of context as a participant in a manuscript on a different topic before, I couldn't help but hope those participants will not ever be able to identify themselves in your manuscript (and that they attend trainings by folx with lived experience).

Line 214, please make sure it says MHP, not MPH.

I appreciate this work and otherwise found the manuscript compelling and well-written. Thank you for doing this study!

7. PLOS authors have the option to publish the peer review history of their article (what does this mean?). If published, this will include your full peer review and any attached files.

Reviewer #1: No

Reviewer #2: No

Reviewer #3: No

---

## [Author Response · Author response to Decision Letter 1]

17 Mar 2022

Author Response: Many thanks for your feedback. Our responses to your thoughtful comments and suggestions are below. 

First, it must be said that the qualitative methodology is very well used in terms of description of the procedures and the reporting of the results.

My concern is with the conclusions. I feel that the first two paragraphs of the conclusion should be followed by the last paragraph of the Continuum section. 

Author Response: We have changed the manuscript based on your helpful comments. We moved part of the last paragraph from the Continuum section to an independent Limitations section (based on recommendations by Reviewer #1) and we moved the second part of that paragraph to follow the first two paragraphs of the conclusion. 

The last 3 paragraphs of the conclusion slightly extrapolate the objectives and results of the study and seem to me to make strong statements about possible consequences for the collective and individual health of trans people, but which go beyond the present article, which is exploratory in nature.

These paragraphs and also the conclusion of the abstract are categorical as if they came from a research of a descriptive/quantitative nature and deserve more nuance.

Author Response: We appreciate this insight, and have revised our conclusion to more carefully reflect the findings of our study, and the inability to generalize from our study to the general “provider” population. We hope that the more nuanced phrasing captures both the nature of our findings, the insights and limitations of qualitative research, and potential implications of our findings on the field at large. 

Furthermore, I feel that the discussion and conclusion should focus on the US context as this issue is more acute in the context in which the research was situated and is present in a different way in other parts of the globe.

Author Response: We added an acknowledgement of the local context to the limitations section (“while the WPATH guidelines are international, our study was limited to providers in the United States, and may not reflect practice patterns in other countries.”), as well as explicitly stating it in the abstract.

Reviewers' comments:

Reviewer's Responses to Questions

Comments to the Author

1. If the authors have adequately addressed your comments raised in a previous round of review and you feel that this manuscript is now acceptable for publication, you may indicate that here to bypass the “Comments to the Author” section, enter your conflict of interest statement in the “Confidential to Editor” section, and submit your "Accept" recommendation.

Reviewer #1: (No Response)

Reviewer #2: All comments have been addressed

Reviewer #3: All comments have been addressed

2. Is the manuscript technically sound, and do the data support the conclusions?

Reviewer #1: Partly

Reviewer #2: Yes

Reviewer #3: Yes

3. Has the statistical analysis been performed appropriately and rigorously? 

Reviewer #1: N/A

Reviewer #2: Yes

Reviewer #3: Yes

4. Have the authors made all data underlying the findings in their manuscript fully available?

Reviewer #1: No

Reviewer #2: Yes

Reviewer #3: Yes

5. Is the manuscript presented in an intelligible fashion and written in standard English?

Reviewer #1: Yes

Reviewer #2: Yes

Reviewer #3: No

6. Review Comments to the Author

Reviewer #1: Thank you for the opportunity to read the revised version of the manuscript. It is a much stronger, coherent, and well-written paper. There are minor, albeit fairly fundamental, revisions I would suggest making.

1) There remains conceptual murkiness around the fundamental concept discussed in this paper - gender identity - or maybe just identity itself. First, the last sentence of the intro where "transgender" is defined is not useful at all. It's circular and also just incorrect. What are non-cisgender identities? For example, trans AND cis adults alike identify as men or as women. Please provide a more nuanced and careful definition.

Author Response: We agree with the reviewer that trans and cis individuals can identify as men or women. We intend this sentence not as a definition, which we believe at this point is basic and familiar enough to the average reader of this paper as to not require a definition, but rather as a clarification that we use the term in a non-exclusive way. We have added clarification that we use the term as an umbrella term, including non-binary and genderqueer individuals among others. 

Relatedly, on the next page, in methods, some clarifications may be helpful. I'm not sure what the reflexivity section is trying to do. The second sentence is very confusing - either say your team consists of "cisgender or non-binary as well as straight and queer" individuals or say it consists of "a diversity of genders and sexualities". As it currently stands I'm not sure what you are trying to say about cisgender, queer, and non-binary people on your team that is unique from the diversity of sexualities and genders. Is queer a gender? Are non-binary people asexual? Are cisgender people straight? Why have those three been singled out from the diversity? Further there seems to be a sentence missing about why it's important to note that no one on the author team has ANY experience with seeking medical transition and how that may impact the findings in this paper. Stating your privilege as researchers would remain true had you included a co-author with this lived experience.

Author Response: 

We have expanded on our experiences with medical transition: 

“While some of us have sought a variety of forms of medical gender affirmation (in addition to the casual, everyday gender affirmation that cisgender people receive by society at large), none of us have accessed gender-affirming hormones. This latter fact may well have impacted our understanding of the process and our interpretations.”

Given our specific identities, our reflexivity statement as it currently stands is the most precise and concise way to share with the reader how our identities might affect our perspectives. To answer the reader’s specific questions, for some of us Queer is a gender identity, for others it is a sexual identity. Regardless of our identities along the cis/trans spectrum, some identify as men, some as women (and some, nonbinary or queer) – thus the diversity of genders is distinct from whether or not we are cis or trans. We found it important to share the diversity of identities in relation to cis/transness in this context, independent of our genders (man, woman, non-binary, etc.). Those specific identities mentioned were singled out, because those are our identities. 

Presumably the reader knows that non-binary people can have any sexuality, as can cisgender people. Agreed that stating our privilege is true regardless of our gendered lived experiences. 

In the next paragraph, purposive sampling was used to recruit providers who "self-identify" as using WPATH or IC. That term is really confusing first of all. To my knowledge, providers do not conflate their treatment model preferences with self-identification. Secondly, it doesn't seem like that inclusion criteria is true. Table 1 says that 17% of your sample uses "other" as their treatment model/guidance. Please align your methods and inclusion criteria with your results.

Author Response: We replaced the words self-identify here with self- described. Our purposive sampling was meant to ensure that we have representation of providers from both groups; identifying as “other” was not an exclusion criterion. Hence both statements (in the introduction and in the table) are correct. 

2) Page 12/line 188-190 there is a redundant theme stated about "broader attitudes regarding gender identity" with very different quotes as examples that follow this same theme - please clarify what is different about these same themes by way of these quotes.

Author Response: Thank you for noticing this error – this was a redundancy in text that was corrected

3) Transsexual is spelled differently in different parts of the paper. Technically, it has two "s"s but some adhere to "transexual" (i.e. not unlike nonbinary vs. non-binary). Please choose which version you are meaning to use and stick with it.

Author Response: We use the term “transsexual” only in citation of other sources, and follow the spelling in the titles as published. We have corrected the spelling where it appears in the body of the text referring to DSM-III. 

4) The quote from #16/line 391 should be modified or commented upon. Why did you think it is important to include a quote, referred to as "their insight," where an ostensible ally is saying that many seem to think this provider has "had more of the Kool-Aid" because they do not gatekeep? Please interpret or share the importance with the reader or don't include it. As it stands, inclusion of this analogy perpetuates the idea of a growing trans medical industrial complex as driven by a cult like that of the Jim Jones kind where it ends in mass suicide for mostly black Americans. Do you think the provider's ideas have "evolved" as you say prior to the quote? I think they sound ambivalent/conflicted about their choices and/or unfairly persecuted by acting as an ally. Maybe say more about what is going on here; otherwise I would not include the "Kool-Aid" comment as an insight around an evolution of ideas.

Author Response: Thank you for the encouragement to further explain this quote (especially in light of the participant’s use of an expression with violent origins and negative contemporary connotations). We have added the following interpretation of the quote: 

“This provider describes their own learning process, with increased understanding of gender as a continuous, social construct. Their newfound understanding, which they acknowledge others might view as ideologic following, has encouraged them to move their practice away from the WPATH model and toward IC.” 

5) Granted i have not had time to do as careful a read of the discussion as I'd like, it seems much stronger and coherent and comprehensive than the initial version (the "Continuum" subheader may need something more, and you might add a "Limitations" subheader which currently falls under "Continuum". Thank you so much for the thoughtful revision.

Author Response: Thank you so much for all your helpful comments. 

We have added a Limitations paragraph to the end of the Discussion section. While more detail could be added to the Continuum section, we agree with the reviewer comments from the first R&R, that this is not the most important finding of the paper. Thus, we have elected not to expand on this further so as not to distract from the main purposes of the manuscript. 

Reviewer #2: (No Response)

Reviewer #3: This is an important topic, and I appreciate what your qualitative study is able to add to the discussion around MHP letter requirements before initiation of gender-affirming care.

I said "no" for number 5 because some of the additions made after the first round of reviews disrupted the flow of the manuscript.

While commenting on and contextualizing the quotes is important, the placement of your critique immediately after the quotes in the results (for example, in lines 229-232) made me feel the opinion of the authors in a way that was distracting as a reader - even though I agree with your interpretation/critique/additions.

Author Response: Thank you for pointing out the excessive opinion here, rather than straightforward interpretation. We have revised the interpretation following the quote here to state “Yet, there is an apparent discrepancy between that claim and the provider’s insistence on having the patient’s identity verified by an MHP. Despite their stated agreement with the principle of believing a patient about their identity, ultimately, it is disbelief or doubt in the patient’s transness that determines this provider’s course of action” 

To back up your critiques of the quotes of the respondents who required letters, you included many more citations in this draft, which was important. Yet again, I think that discussing the quote within the context of available literature (for example, lines 275-285) would have been more appropriate in the discussion section.

Author Response: Agreed! We moved this section to the discussion, which hopefully also improves the flow. 

In fact, I found myself wondering who providers 13-15 were, and getting upset that they feel qualified to do gender-affirming care. Perhaps this was the intent, but it made it difficult for me to maintain attention through the rest of the manuscript. And, as someone who has been quoted out of context as a participant in a manuscript on a different topic before, I couldn't help but hope those participants will not ever be able to identify themselves in your manuscript (and that they attend trainings by folx with lived experience).

Author Response: Sadly, we agree. We understand that they may be upset if they recognize themselves in the manuscript, and may feel misunderstood; it is our interpretation that they do not necessarily have full awareness of the meaning of some of their statements, hence may feel misquoted. Nevertheless, having discussed this within the research team extensively, we ultimately feel that these are the best interpretations of our data, and that we are giving an honest account of some of the issues in the field

Line 214, please make sure it says MHP, not MPH.

Author Response: Corrected, thank you.

I appreciate this work and otherwise found the manuscript compelling and well-written. Thank you for doing this study!

7. PLOS authors have the option to publish the peer review history of their article (what does this mean?). If published, this will include your full peer review and any attached files.

Do you want your identity to be public for this peer review? For information about this choice, including consent withdrawal, please see our Privacy Policy.

Reviewer #1: No

Reviewer #2: No

Reviewer #3: No

---

## [Editor Report · Decision Letter 2]

8 Jul 2022

Initiating gender-affirming hormones for transgender and non-binary people: A qualitative study of providers’ perspectives on requiring mental health evaluations

PONE-D-20-30031R2

Dear Dr. Stroumsa,

We’re pleased to inform you that your manuscript has been judged scientifically suitable for publication and will be formally accepted for publication once it meets all outstanding technical requirements.

Kind regards,

Vanessa Carels

Staff Editor

PLOS ONE
---

## [Editor Report · Acceptance letter]

10 Aug 2022

PONE-D-20-30031R2 

Initiating gender-affirming hormones for transgender and non-binary people: A qualitative study of providers’ perspectives on requiring mental health evaluations 

Dear Dr. Stroumsa:

I'm pleased to inform you that your manuscript has been deemed suitable for publication in PLOS ONE. Congratulations! Your manuscript is now with our production department. 

Kind regards, 

on behalf of

Dr. Vanessa Carels 

Staff Editor

PLOS ONE